# Adapting to ulcerative colitis to try to live a 'normal' life: a qualitative study of patients' experiences in the Midlands region of England

Christel McMullan,[1] Thomas D Pinkney,[2] Laura L Jones,[1] Laura Magill,[3] Dmitri Nepogodiev,[2] Shri Pathmakanthan,[4] Rachel Cooney,[4] Jonathan M Mathers[1]

[1]Institute of Applied Health Research,University of Birmingham, Birmingham, UK
[2]Academic Department of Surgery, University of Birmingham, Birmingham, UK
[3]Birmingham Clinical Trials Unit, University of Birmingham, Birmingham, UK
[4]Department of Gastroenterology, University Hospitals Birmingham NHS Foundation Trust, Birmingham, UK

**Correspondence to**
Dr. Jonathan M Mathers;
j.m.mathers@bham.ac.uk

## ABSTRACT

**Objective** To provide a framework that is able to categorise whether patients are able to adapt to and lead a 'normal' life with ulcerative colitis (UC) and to detail the factors that influence this.

**Design** Qualitative research study using in-depth semi-structured interviews.

**Setting** Four clinical sites in the West and East Midlands regions of England.

**Participants** 28 adult patients diagnosed with UC for years between 1 and 22.

**Results** Medication was rarely sufficient for patients to adapt to UC and live as 'normal' a life as possible. Virtually all patients tested and adopted non-medical adaptation methods to improve physical and psychological well-being, to help them carry on working and to prevent embarrassment. In addition, some patients benefited from outside support providing them with practical, emotional and/or financial help. In conjunction with adaptation strategies and the time to adapt, this meant that some patients with severe clinical disease were able to maintain a sense of normality in life. Patients reported that clinicians were not always receptive to discussion of the broader context of life with UC.

**Conclusions** Patients' experience of UC and their ability to adapt in order to maintain a sense of normality in life is a complex interplay of symptoms, adaptation strategies and outside support. Over time patients test out a variety of non-medical adaptation strategies. Awareness of this may help clinicians and researchers to understand patients' views on the role of medical and other therapies. Further research around the utility of this framework in clinical practice and research is now required.

**Trial registration number** ISRCTN56523019, results.

## INTRODUCTION

Ulcerative colitis (UC) is a form of chronic inflammatory bowel disease (IBD), which affects the large bowel. Over 1 45 000 people in the UK[1] and 9 00 000 people in the USA[2] are living with UC. The condition is most prevalent in Northern Europe and North America, although the number of people affected by UC is increasing recently in developing

### Strengths and limitations of this study

► This study is an in-depth participant-focused qualitative study providing rich and detailed accounts of patients' experiences of living with and adapting to ulcerative colitis (UC).

► This is the first detailed analysis that has attempted to categorise whether patients are able to adapt to and lead a 'normal' life with UC and also to detail the factors that influence this.

► The semi-structured interviews provided patients the opportunity to talk about their experience to someone outside of their clinical team.

► The research is cross-sectional in nature, limiting the potential to observe the temporal components of adaptation over time.

► The patients interviewed were participating in a pilot trial, which means that the range of patients was influenced by the trial eligibility criteria.

countries.[3] There is also an increasing incidence of UC among younger patients.[4] Around 40% of patients will experience a relapse annually, with between 20% and 30% requiring total colectomy in their lifetime.[5 6] Reducing relapse rates and disease progression is a priority for patients.[7]Ulcerative colitis has been shown to have a significant negative impact on health-related quality of life (HRQoL).[8]While many patients' lives are severely disrupted as, for example, the disease affects their ability to work as well as their social and family life,[8–11] these impacts are not always directly correlated, from a clinical perspective, with disease activity and severity of symptoms.[12]Patients with mild clinical disease can experience significant impacts on HRQoL and psychological morbidity, while some with apparently severe disease activity seem to 'cope' relatively well.

Commentators have argued that often the focus of the medical management of IBD is too much on acute episodes, not necessarily

reflecting the chronicity of IBD and therefore the overall patient experience of disease over time.[13] Understanding more about living with IBD can help to highlight the role that medical and surgical management plays. The patient's perspective on living with UC may sometimes be difficult for clinicians to interpret as they may underestimate or overlook the overall effects of disease on patients' lives.[8]

Qualitative research is well placed to describe and understand patients' views on disease and treatment.[14] Within IBD and UC, there is a small but informative body of qualitative research relevant to these issues and work to date has described patients' perspectives on the impact of IBD; of the physical symptoms and broader issues, for example, on work, lifestyle, daily activities, social life and psychological well-being.[8–11 15–20] Research has suggested that patients are engaged in a 'fight' for some sort of health-related normality and that achieving a 'new normal' is their ultimate goal.[15] Trying to live as 'normal' a life as possible is something that patients reflect on when they talk about life with IBD.[12] This body of research describes patients' views on how their lives are affected by IBD and shows that adaptation (ie, a desire for normality) is an overriding concern. However, to date nobody has provided a clear framework to help understand normality and adaptation in UC.

The aim of this paper is therefore twofold: (1) to provide a framework that is able to categorise whether patients are able to adapt to and lead a 'normal' life with UC and (2) to detail the factors that influence this.

## METHODS
### Setting and study design
Participants were recruited from four hospitals in the West and East Midland regions of England and were taking part in the National Institute for Health Research (NIHR) for Patient Benefit-funded ACCURE-UK trial.[21] This randomised external pilot trial was exploring the feasibility of a multi-centre randomised controlled trial (RCT) of therapeutic appendicectomy for the treatment of UC (intervention arm) in addition to standardised medical therapy (control arm). The trial included integrated qualitative research, the main aim of which was to investigate the acceptability of the trial and understand this within the context of patients' everyday lives with UC.

The theoretical underpinnings of our approach most closely align with interpretive description. Originally described in nursing and drawing on methods from established qualitative methodologies, this approach has a heavy emphasis on understanding and informing clinical practice.[22 23] The research methods we describe (eg, for interviews—broad purposive sampling, iterative in-depth data collection and inductive analysis without a priori theoretical tools) are concordant with a generic interpretive approach to addressing our research aims.

### Sampling and recruitment
We recruited a diverse sample (age, gender, time since diagnosis) of patients from both arms of the pilot RCT. Clinical staff initially discussed the qualitative research with patients. With consent, the contact details of patients expressing an interest were passed to the researcher conducting the interviews (CM), who then took informed consent prior to interview.

### Data collection
Semi-structured one-to-one interviews were conducted by CM between randomisation and surgery for those allocated to intervention arm, and shortly after randomisation for those in the control arm. CM is a non-clinical trained qualitative researcher independent from participants' clinical care and the day-to-day trial management. This was clearly communicated to participants. Participants were asked to select a convenient time and place for interview. Most chose to be interviewed at home, although a small number of interviews were conducted on hospital or university premises, or by telephone.

The interview schedule was informed by the existing literature and consultation with the wider research team. It included discussion of patients' views and experience of life with UC since initial diagnosis: their symptoms, their flare ups and their impacts, their medical treatment and their perspectives on adaptation to UC. In addition, a section of the interview focused on participants' perspectives on the RCT and trial processes (data not reported here). Interviews were conducted in a participant-focused open-ended manner. After initial piloting, data collection and analysis took place iteratively. This continued until the research team judged that the data and sample had sufficient depth and breadth to address initial research questions.[24] Field notes were kept after the interviews to record factors that might have influenced the conduct and the analysis of the interviews.

### Data analysis
All interviews were audio-recorded and transcribed verbatim by a specialist external transcription company. Data were analysed thematically and managed using Computer-Aided Qualitative Data Analysis Software (CAQDAS), NVivo10. The analysis was informed by the framework analytical approach.[25] First, interview transcripts were reviewed several times and open coding was undertaken. These codes were then reviewed and categorised and the dataset was indexed. Categories were refined into overarching themes from descriptive accounts of the data. Final analysis and explanation was facilitated by use of a typology and associative analysis using charting. The analytic typology is described in more detail below. A sample of interviews were coded by an independent researcher during initial coding. The research team ensured not only that code saturation was reached but also that we had a deep and rich understanding of what the themes derived during analysis were about.[26] The final analysis and interpretation were discussed among

| Table 1 | Normality types. |
|---------|------------------|
| **Type/category** | **Definition and constituent elements** |
| Non-normal life | ▶ Some people may find it difficult to accept their condition and attempt to carry on as before, even hiding their condition from others. This may have negative consequences when this is not successful<br>▶ Symptoms may be devastatingly intrusive and unmanageable for some. It may be difficult or impossible to carry out day-to-day activities, for example, because of symptoms and/or embarrassment, despite acknowledging and trying to adapt to the condition<br>▶ The condition can threaten peoples' self-identity and associated sense of normality<br>▶ There may be a feeling of loss of control over the disease and life |
| Normal life | ▶ For some, medication may be so effective that they can carry on as normal that is, as before the diagnosis<br>▶ Other patients may experience mild symptoms with minimal impact on life, daily activities and self-identity, with control facilitated by medication<br>▶ Some patients with more severe symptoms may reconceptualise normal life by including their condition and its impacts in a life that accommodates the disease. This will include acceptance of the condition and reorientation of self-identity and thoughts about what constitutes a healthy body |

Adapted from Sanderson *et al*.[27]

the research team, and with one patient who took part in the interviews.

### Analytic typology

During analysis, we have applied a typology devised from qualitative data gathered with patients diagnosed with rheumatoid arthritis, another chronic inflammatory disease of relapsing remitting nature.[27] A typology is a classification system that consists of categories that describe different types of a phenomenon of interest, here adaptation to life with UC and the ability to attain a new sense of normality. Typologies are sometimes used to categorise observations in qualitative data.[28]

In addressing our first research aim, we have categorised participants into two types (see table 1 and table 2), one of broadly positive experience and adaptation to UC (the normal life) and one of negative experience and non-adaptation (the non-normal life). This typology encompasses patients' reactions to their illness, their attempts to adapt to it and success or failure in this. Of note, these are not static categorisations; patients may move between them, over time and as a consequence of changing circumstances. We have used the accounts given by participants during the interviews to assign them to these two broad categories: at the time of the interview and at the time points since diagnosis.

### RESULTS

The following analysis is based on data from interviews with 28 patients (table 2), which lasted between 22 and 58 min; 16 women and 12 men aged between 18 and 57; 24 of white and 4 of Asian ethnic origin; diagnosed with UC between 1 and 22 years. All were in remission at the time of interview but had experienced a disease relapse within the preceding 12 months (RCT eligibility criteria). All but one were on medical therapy, with three having taken biologic therapies, such as infliximab and adalimumab. One patient had decided to discontinue all

medical therapy due to side effects. None had undergone any colonic surgery previously. There was no withdrawal.

Findings are presented as follows:
▶ the normality types experienced and described by our participants;
▶ their adaptation strategies and intended purpose;
▶ the outside support to adaptation;
▶ how these factors interact with symptoms to influence the ability of a patient with UC to lead a normal life.

### Participants' descriptions of normal and non-normal life since diagnosis

Our analysis of patients' accounts suggests that at the point of interview, 12 of the 28 participants in this sample could be categorised as experiencing some form of normal life with UC. There was no clear relationship between the length of time since diagnosis and the type of normality they were experiencing at the time of the interview. At diagnosis, all participants had described a period of non-normal life with significant disruption due to disease activity and symptoms. Some also talked about a need to understand and acknowledge the significance of the diagnosis itself. Participants suffered from intrusive symptoms (stomach cramps, bleeding, more frequent bowel movements); and in many cases, it took several months to receive a definitive diagnosis and treatment, with several patients ending up being admitted to hospital. As a result, they had to take time off work and were not able to carry on with their daily activities:

> Diagnosed in October. We were actually away in Spain when I had the flare up, and I was in hospital in Spain for a week, and they did all the tests there, then when I got back to England they did it all again in October. I was due to go in for the camera, and when I got there they said that I'd got so weak that they wouldn't be able to do it, but they kept me in, and obviously I was

**Table 2** Sample characteristics including normality types, adaptation strategies and outside support

| Patient ID | Age | Year of diagnosis | Marital status | Employment status | Disease severity (mild, moderate or severe)* | Current medication regiment† | Dominant type of normality at interview | Previous period of normality described (non-normal life only)‡ | Adaptation strategies used‡ | Outside support described‡ |
|---|---|---|---|---|---|---|---|---|---|---|
| 1 | 31–40 | 2004 | Married | Self-employed | Moderate/severe | 5-ASA, immunomodulators | Non-normal | N | Y | N |
| 2 | 31–40 | 2013 | Married | Employed | Mild | 5-ASA, immunomodulators | Non-normal | N | Y | N |
| 3 | 31–40 | 2003 | Married | Employed | Moderate | 5-ASA, immunomodulators | Non-normal | Y | Y | Y |
| 4 | 21–30 | 2014 | Cohabiting | Employed | Mild | 5-ASA, immunomodulators | Non-normal | N | Y | Y |
| 5 | 51–60 | 2009 | Married | Not employed | Moderate | 5-ASA, immunomodulators | Non-normal | N | Y | Y |
| 6 | 31–40 | 2013 | Married | Employed | Moderate/Severe | 5-ASA, immunomodulators | Non-normal | N | Y | Y |
| 7 | 41–50 | 2012 | Single | Self-employed | Mild | 5-ASA, | Non-normal | N | Y | N |
| 8 | 21–30 | 2009 | Single | Employed | Moderate | 5-ASA, immunomodulators | Non-normal | N | Y | Y |
| 9 | 21–30 | 2011 | Single | Employed | Mild | 5-ASA, biologics | Normal | N/A | Y | Y |
| 10 | 51–60 | 2014 | Divorced | Self-employed | Moderate | 5-ASA | Non-normal | N | Y | N |
| 11 | 31–40 | 2002 | Single | Employed | Mild | 5-ASA | Non-normal | Y | Y | N |
| 12 | 41–50 | 2013 | Divorced | Self-employed | Mild | 5-ASA | Normal | N/A | Y | N |
| 13 | 21–30 | 2013 | Single | Employed | Mild | 5-ASA | Normal | N/A | Y | Y |
| 14 | 21–30 | 2014 | Single | Employed | Moderate | Immonumodulators, biologics | Non-normal | N | Y | N |
| 15 | 31–40 | 1995 | Single | Employed | Moderate | 5-ASA, immunomodulators | Normal | N/A | Y | Y |
| 16 | 21–30 | 2014 | Single | Employed | Moderate | 5-ASA, immunomodulators | Non-normal | N | Y | Y |
| 17 | 31–40 | 2006 | Single | Employed | Mild | 5-ASA, immunomodulators | Normal | N/A | Y | N |
| 18 | 51–60 | 2014 | Separated | Not known | Mild | 5-ASA | Normal | N/A | Y | N |
| 19 | 41–50 | 2003 | Married | Employed | Moderate | 5-ASA, | Normal | N/A | Y | Y |
| 20 | <20 | 2014 | Single | Unemployed | Mild/moderate | 5-ASA | Non-normal | N | Y (medication only) | Y |
| 21 | 41–50 | 2012 | Married | Self-employed | Moderate/severe | 5-ASA, immunomodulators | Non-normal | N | Y | Y |
| 22 | 41–50 | 1993 | Married | Employed | Mild | 5-ASA | Normal | N/A | Y | N |
| 23 | 21–30 | 2014 | Cohabiting | Employed | Mild | 5-ASA | Normal | N/A | Y (medication only) | Y |
| 24 | 41–50 | 2003 | Married | Employed | Mild | None | Normal | N/A | Y | Y |
| 25 | 41–40 | 2013 | Married | Employed | Mild | 5-ASA, immunomodulators | Non-normal | N | Y | N |
| 26 | 51–60 | 2004 | Married | Employed | Moderate | 5-ASA, immunomodulators | Normal | N/A | Y | Y |
| 27 | 31–40 | 2011 | Married | Employed | Mild/moderate | 5-ASA, immunomodulators, biologics | Normal | N/A | Y | Y |
| 28 | 31–40 | 2013 | Married | Employed | Moderate | 5-ASA, immunomodulators | Non-normal | N | Y | N |

Continued

**Table 2** Continued

| Patient ID | Age | Year of diagnosis | Marital status | Employment status | Disease severity (mild, moderate or severe)* | Current medication regiment† | Dominant type of normality at interview | Previous period of normality described (non-normal life only)‡ | Adaptation strategies used‡ | Outside support described‡ |
|---|---|---|---|---|---|---|---|---|---|---|

*Disease severity was categorised based on patients' descriptions of their symptoms and through analysis of this and descriptions of the impacts of ulcerative colitis that patients provided for example, the amount of time patients took off work, and the number of flare ups patients described.
†Categorical data on type of medication at the time of interview.
‡Y =yes, N=no.
ASA, 5-aminosalicylic acid; N/A, not applicable.

in hospital then for three weeks, or nearly four weeks I believe it was at the time (Patient 18)

Following an initial period of disruption and non-normal life around the time of diagnosis participants' accounts vary, with some seemingly not having attained or regained any sense of a normal life. Others had managed to do so, some temporarily. Only three patients described a period where they felt that they had been relatively unaffected as their symptoms were largely controlled by medication, such that they were able to carry on life without much perceived impact. For example, one male participant who was diagnosed in 2002 talked about not having a flare up while managing UC for the first 5 years with medication. None of the participants indicated that medication had dealt with all of their symptoms to the extent that life was the same as before the onset of disease.

### Participants' descriptions of adaptation strategies and their intended purpose

During the interviews, all participants described attempts to adapt to their condition, and all but two participants actively tested or adopted approaches that were additional to medical therapies. We have categorised the adaptation strategies described according to their intended purpose. Most commonly, participants aimed to achieve four things (table 3):

1. To improve physical well-being
2. To improve psychological well-being
3. To carry on working
4. To prevent embarrassment

#### Improve physical well-being

This category describes participants' strategies to reduce physical symptoms and the frequency of UC flare ups, and also to do so by addressing perceived triggers of flare ups. The physical symptoms were a very important part of the great majority of patients' accounts, as they made them extremely fatigued:

Just draining, not tired, not sleepy, just drained. [Name] knows what I'm like, by the time I've been five times and I've just got no energy, just no energy to do anything, and I just the best thing is just to give into it. I say to [name], you're going to have to count me out for the rest of the day, and just give into it (Patient 21)

The most common adaptation approach in this category was medication, with all but one participant taking regular medication to improve physical well-being. The one participant, who had ceased taking medication due to side effects, reported that smoking helped to reduce symptoms. During the interviews, it was clear that there were varied perspectives on whether medication was 'working'. Some reported that medication was effective and others that it was not. Some participants were unsure as to how much it helped:

| Table 3 | Adaptation strategies and aims* |
|---|---|
| **Aims of adaptation** | **Adaptation strategies** |
| Improve physical well-being (reduce symptoms and flare ups; prevent triggers of flare ups) | ► Medication<br>► Change diet<br>► Probiotic use<br>► Complementary therapies (eg, acupuncture, hypnotherapy)<br>► Finding a balance between doing too much and too little (pacing)<br>► Exercise<br>► Yoga<br>► Relaxation classes<br>► Smoking<br>► Personal research (including forums, Crohn's and Colitis UK, social media) |
| Improving psychological well-being | ► Positive and proactive attitude<br>► Fighting on/soldiering on in activities (eg, playing netball, going on holiday)<br>► Finding a balance between doing too much and too little (pacing)<br>► Mindfulness courses<br>► Yoga<br>► Relaxation classes<br>► Personal research (including forums, Crohn's and Colitis UK, social media) |
| Carry on working | ► Reducing working hours; selling off part of own business<br>► Rearranging working patterns<br>► Changing jobs<br>► Getting up earlier<br>► Finding a balance between doing too much and too little (pacing)<br>► Personal research (including forums, Crohn's and Colitis UK, social media) |
| Prevent embarrassment | ► Situational avoidance (not going out; avoiding certain places)<br>► Planning (timing, location, duration of activities)<br>► Wearing nappies<br>► Personal research (including forums, Crohn's and Colitis UK, social media) |

*A number of adaptation methods appear across several categories.

I don't notice any effect with the anti-inflammatories (…) It generally doesn't keep it [UC] away, because like I say I can be fully recovered, go nine/ten months of the year whilst taking anti-inflammatories every day, and still go into a flare up. But whether if I didn't take them I don't know if I'd go into a flare up quicker. So I don't know (Patient 1)

Another common adaptation approach to improve physical well-being was to change diet. As with other adaptation approaches, this was often a trial and error strategy with the aim to establish which foods tended to worsen symptoms or trigger flares. Some people reported keeping a food diary for several weeks or months to identify patterns in their food intake and symptoms:

When I came out of hospital for about a month I kept a list of all the food I'd eaten every day and what I'd drank, so that if I went into a flare up I could look back and say maybe it was that, so I'll try it one more time and if it happens again I'll not eat or drink that again (Patient 10)

Other adaptation strategies in this category include the use of complementary therapies (eg, acupuncture or hypnotherapy), exercise, yoga, relaxation courses to avoid triggers (stress) and the pacing of activities to find a balance between symptoms and the ability to do things.

### Improve psychological well-being
When discussing issues pertaining to psychological well-being, several participants stated that they have a positive and proactive outlook on life with UC. We believe that this signals acceptance that the disease was not going to go away, and describes an associated attitude and approach to living with it. This was sometimes evident in discussions about the need to 'soldier on' and maintain some or all of the activities that participants pursued prior to diagnosis, such as playing sports, going out and going on holiday. Often, similar to other adaptations, this would involve a process of testing the boundaries of what was possible and finding a balance between the desire to keep going and the potential negative (eg, physical) consequences of doing too much. For example, Patient 19, a 42-year-old woman first diagnosed in 2003 described how, over the years, she adapted her activities to have more control of her condition, but was keen not to be a 'victim', which she likened to being 'too depressive':

I just refuse to give up and not do anything, and I know that the exercise is good for me. But equally

I know that doing too much exercise is bad for me, because it will make me ill and run down (…) refusing to not do anything and be a victim and just sit at home doing nothing, because that's just too depressing to even think about isn't it? (Patient 19)

Specific activities that were adopted by participants in an attempt to improve psychological well-being included attending mindfulness and relaxation courses, and also yoga.

## Carry on working

Being able to maintain a working life was expressed as a key concern by the majority of our participants. The main strategy in order to carry on working was reducing working hours, because of fatigue (one of the most commonly reported physical symptoms), a lack of concentration and the need to constantly be near toilets. Spending less time in work meant that participants had more time for themselves to rest at home:

There's certain things I can't do at work, like I don't do playground duty, because that way I get an extra break, I don't teach PE, it just wears me out really, it is quite a physical job. I've had to reduce my days as well, I did work three days a week before I was diagnosed, and now I only work two (Patient 6)

Participants who reduced their working hours found themselves to be more efficient at work. However, this had a financial impact, placing burden on participants and their families.

Other strategies to maintain working included rearranging working patterns (eg, working only in the afternoon as symptoms were worse in the morning; spreading working hours throughout the week), changing jobs to something less demanding, and getting up earlier in the morning to have more time to deal with the symptoms and be ready in time for work.

## Prevent embarrassment

Avoiding and preventing the embarrassment potentially caused by UC was a key theme in the data. This very often restricted participants in their daily life, for example, with the fear of being embarrassed by not being able to find toilets when needed:

I don't go out as much as I did before, and when I go out I'm very nervous, first thing I do a check where is the nearest toilet even if I don't need it, and some days if my friends want to go out or do something I might be too fatigued (Patient 4)

One strategy employed to prevent embarrassment was situational avoidance with some participants reporting not going out or not going to places where toilets are not readily available. While such avoidance may be perceived as being effective in preventing embarrassment, in turn it could result in feelings of isolation:

Well because I've been having so many flare ups and been unwell a lot I don't have so much of a social life now, and every time I try and plan things in I end up being ill a lot and having to cancel all the time, so I stopped making forward plans now to be honest, because if there are things a few months away I just say to friends that I have to let you know at the time, because I never know what my health is going to be like. So it has had quite an impact socially. It can be a bit isolating as well, that's a bit difficult (Patient 8)

Other participants discussed having to carefully plan, such as for nights out, around the location of the toilets, to limit the frequency of their social outings, or to try to plan them during remission periods. Finally, one patient reported being able to go out because he was wearing nappies, which meant he was not as restricted by the location of toilets.

Personal research, including participation in disease-specific internet forums and social media, was described as contributing to each of the aims associated with adaptation strategies that were tested by patients:

It's just a Facebook group of everybody who has got Crohn's and colitis really, it's a group where you could obviously speak to other people about your condition. There's quite a lot of support on there, and it has been mentioned, and so there's obviously people already know already. There seems to be a lot of support for anything, awareness of disease, and look at ways of improving the condition (Patient 3)

## Outside support to adaptation

Some patients spoke about the influence of outside support in enabling life and adaptation to UC. Such support was not necessarily asked for or actively sought by participants but nevertheless helped them live and manage UC on a day-to-day basis. They provided an invaluable source of social support for several participants.

The main support evident in the data were as follows:

► Help from parents/close family members (practical, emotional and financial)
► Help from a partner and/or children (practical and emotional)
► Help from friends (practical and emotional)
► Help from employer/workplace (practical)

Support from parents and other close family members included moving closer to participants or having them live with them in order to help with day-to-day tasks, for example, the school run, household chores and give them financial help:

My parents have moved house to be nearer (…) But they rushed the move because I was poorly, they actually moved while I was in hospital, because while I was in hospital they were staying here for half the week to have my boys, because when I was in hospital both my boys were preschool age, they were two

and four, so obviously someone had to have them. (Patient 6)

Participants also often discussed the practical and emotional support provided by their partners and children:

Luckily my daughter is quite mature for her age, because she's only 11, but she does understand, and she's quite good, and she will help me, and if I need help she is always there, and make me a cup of tea and little things like that. So yes quite lucky in that respect. (Patient 15)

Finally, work managers and the workplace often offered practical support to patients, mainly by providing flexible working hours, helping them to attend hospital appointments or deal with the impact of symptoms:

Work have been pretty good, been really good with allowing me time off for appointments and things. (Patient 23)

One patient described how his workplace had recognised UC as a disability, and that this was invaluable in providing the means to accommodate the impacts of the disease, for example, by taking time off work for hospital appointments or because of symptoms.

### How symptoms, adaptation strategies and outside support interact to influence adaptation and the ability to live a normal life with UC

Whether patients are able to maintain or regain a sense of a normal life with UC appears to be a complex interplay between disease activity and symptoms, the impacts of these, attempts to adapt and outside support (figure 1). Those patients who, overall, describe and judge their disease activity to be mild were also more likely to indicate that they were able to regain a sense of normality. Most, but not all of these patients, thought that their

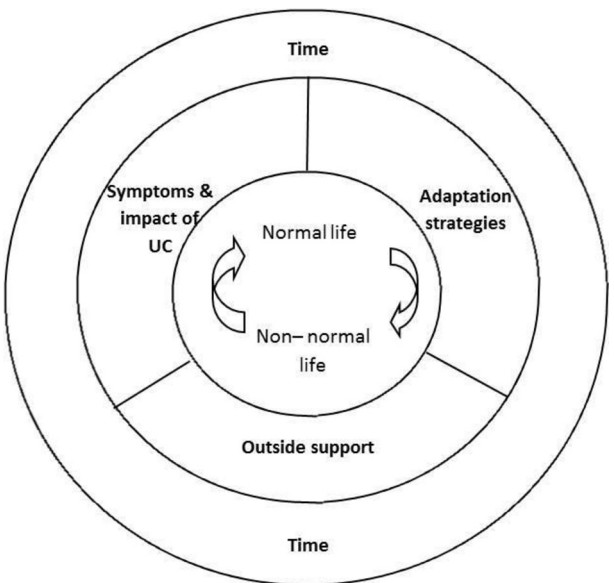

**Figure 1** Influences on normality in ulcerative colitis.

medication was successful in helping them control symptoms and impacts. All but one had adopted some adaptation strategies and half described the positive influence of outside support. However, several patients who described mild disease did not convey a sense of a normal life. Most of these had been diagnosed relatively recently, related uncertainty about the effectiveness of their medication, were testing out adaptation strategies and were less likely to describe the positive influences of outside support.

Those patients that described moderate or severe disease activity, symptoms and impact, also frequently described a non-normal life. Again, several of these patients were recently diagnosed and were less certain about the effectiveness of medication in controlling disease activity and symptoms. Only one of these patients diagnosed over 10 years prior to interview detailed a long period of normality while medication controlled symptoms and prevented flare ups. More recently, with frequent flares, this patient described her life as a 'roller coaster' with periods of severe disruption (flare ups) and intervening remission:

It is a bit like a rollercoaster, there's like it can be relatively stable for a while, but a flare to me is when I start to lose blood and the urgency starts to creep up, and I find it difficult to like I say get out of the house in the morning and I have to use the toilet frequently at work, or if I'm out, and they're specifically my flare up symptoms, and it's usually always with passing blood as well, and like I said probably about three of those where I've been really bad over the past 12 months. (Patient 3)

In contrast, four patients whom we interviewed, despite perceptions of more severe disease and flare ups, talked about life with UC in a manner concordant with regaining a sense of a normality. All but one had lived with UC for over 10 years and while they were unsure about the effectiveness of their medication in controlling symptoms and flare ups, the use of adaptation strategies and descriptions of positive outside support suggested that over time they had been able to adapt to a life accommodating UC. For example, Patient 19 adapted her working hours to be less tired:

I also changed my hours, when I had my [previous/most severe] flare up three years ago, I do 22 hours, and I used to do it in three days, so I would come in at half eight, and eight o'clock in the morning, and I wouldn't finish until half four, I would have half an hour for lunch. After I was last ill it was easy for me to change my hours, and I do the same hours, but instead of doing it in three long days I do four short days, because for me it's just easier, so then when I start to get tired I think well I'm going home mid-afternoon, which is why I finish at half two, because it's easier for me physically to cope with that, and mentally it's much easier. (Patient 19)

Another participant described how his wife had been key in providing emotional and practical support to help with adaptation:

> I think the physical support, having somebody to lean on, you don't realise it at the time when you're on your own that you've just got… obviously got family around you but nobody directly in your life, in your house. I think just that moral and physical support there is a help, every little helps, it's only a small percentage, it's a definite help (Patient 26)

### Time

Participants' accounts and descriptions of attempts to adapt to UC since diagnosis demonstrate that this is an iterative and ongoing process of trial and error. People test the boundaries of abilities and the role of adaptation methods and may take time to gain the necessary facilitators to adaptation, and also to reach a point of acceptance of the disease or not. Similarly, the role of medical therapies in improving physical well-being is often reported as being tested iteratively over time in collaboration with healthcare professionals. Finally, circumstances may change over time, such as the frequency and severity of flares.

### DISCUSSION

To our knowledge, this is the first article in UC and the IBD sphere to propose a framework of normality (figure 1) to categorise and understand patients' overall experience of life with the disease, and the factors that influence adaptation to it. The concept of disease-related normality helps us to think about how disease impacts on patients holistically; the interplay of physical symptoms; the consequences for daily life and how patients see themselves and their lives as a result. It shows how disease can be a disrupting life experience (the 'non-normal life'), and how patients approach adaptation and whether, broadly speaking, they are successful in this (the 'normal-life'). In the absence of cure, disease symptoms, adaptation strategies and outside support cumulate to influence patients' views on life with the disease. This can change over time, with acceptance of the disease and need to adapt, and with changing circumstances, such as treatment, disease progression and the presence or absence of outside support.

For patients with UC to be able to live a 'normal life', traditional medical management is likely to be important but as highlighted in this sample, may be insufficient on its own. The majority of patients described other adaptation methods that they felt were required, that they had tried and tested or were in the process of testing. These methods aimed to achieve different but interrelated things; physical and psychological well-being; the ability to carry on working and to prevent embarrassment. There may not be good evidence that the aims of the adaptation strategies can be achieved by the specific methods adopted by patients. As we have described, patients themselves often actively reflect on the efficacy of their actions, for example, in monitoring and adjusting diet to influence physical well-being. Despite this, the trial and error adoption of different strategies may in itself be important as part of a process of accommodating disease and being active in doing so. Research in diabetes[29] has proposed that such actions are important for young people in attempting to 'master' their disease, and recent research around normality in cancer survivorship[30] has also indicated that the act of doing things is important in its own right regardless of the outcome of the action itself.

It was also clear that adaptation and normality were specific to the individual, meaning different things to different people. Reformulating identity and sense of self with disease is crucial for adaptation. Loss of sense of self has been identified as integral to chronic disease experience generally,[31] and other researchers interested in normality in IBD have also observed that identity influences views on normality.[9 15 17] For a normal life to be achieved, the disease and its impacts have to be accommodated via adaptation, and thereby integrated into a new, and perhaps dynamic, sense of self. We did not talk to anyone with UC who described their life as being the same as before the onset of disease. However, for many, the physical burden of disease will preclude this adaptation and for others there may be a significant period of time before adaptation is achieved.

The in-depth participant-focused qualitative research approach used here has allowed us to gather and analyse rich and detailed accounts of patients' perspectives on living with and adapting to UC. We have talked to a range of patients who have lived with UC for varying periods of time. There are, however, limitations to our approach. All of the patients interviewed were participating in a pilot trial and as such the range of patients was influenced by the trial eligibility criteria. This meant, for example, that everybody was in remission at the time of the interview. Those with long-standing mild and controlled disease were less likely to figure in this research as patients had to have had a relapse within the 12 months preceding recruitment. However, these are potentially the patients most likely to be able to adapt. Additionally, this research is cross-sectional in nature and so we have not been able to observe the temporal components of adaptation over time, although we have interviewed patients who have been diagnosed with UC between 1 and 22 years. Further research with a broader range of patients and perhaps with a longitudinal design may provide further insight.

Several patients commented that they had welcomed the opportunity to talk about their views in-depth to someone outside of the clinical team. A few of them said that they felt that their clinicians were not always receptive to the broader context that they were able to talk about as participants in this research. The implications of this for clinical practice need to be explored

further. Naturally, clinicians may focus on the physical and clinical manifestation of disease. As described here, physical well-being is a core component of adaptation and a 'normal life'. However, physical disease and its impacts are experienced holistically by patients and those around them. The efficacy of medical and surgical management and patients' relationship to this, for example, via adherence, shared decision-making re-treatment options, and interactions with healthcare professionals and services, will be judged by patients within the broader context of adaptation that we have described here. Other authors have recently reflected on the need to identify and address poor adaptation to IBD as a means to impact on quality of life and also potentially the course of disease.[32] The consideration and development of interventions that facilitate adaptation and self-management in UC may help to reflect the holistic experience and priorities of patients with UC.

## CONCLUSION

In this study, we have described the views of patients, who for the most part did not use biologics, on the factors that influence their adaptation to UC and whether they perceived that they are able to regain and maintain a 'normal life'. We have shown that adaptation to UC is complex and that medication alone is most often insufficient to achieve this. Symptoms, adaptation strategies and outside support all have an influence on whether patients manage to regain and maintain normality over time. This holistic view of adaptation to UC will help clinicians and researchers to understand patients' views on life with the disease and the role of medical and other therapies within this.

**Acknowledgements** The authors are grateful to the patients for their participation in this study and the clinical staff at the relevant hospitals for facilitating this research. We also wish to thank study interviewee for kindly agreeing to review and comment on our manuscript. The authors also thank the consultants and the research nurses who facilitated the study: Baljit Singh, Neil Cruikshank, Haney Youssef, Robert Church, Tracy Henn, Mandip Narewal, Alison Moore, Lisa Richardson and Julie Reddan.

**Contributors** All authors made a substantial contribution to the manuscript. JMM designed and conceived the qualitative research with TDP. CM conducted the interviews. Data analysis was carried out by CM, JMM and LLJ, and interpretations checked with all authors. The manuscript was drafted jointly by CM and JMM. All authors revised the manuscript for intellectual content. All authors have read and approved the final manuscript.

**Funding** This work was supported by a grant from the Research for Patient Benefit (RfPB) programme of the National Institute for Health Research (NIHR) (grant no: ISRCTN56523019).

**Competing interests** None declared.

**Patient consent** Obtained.

**Ethics approval** Ethical approval for the ACCURE-UK trial was granted from the North East Tyne and Wear South 200 Research Ethics Committee, REC number 14/NE/1143.

**Provenance and peer review** Not commissioned; externally peer reviewed.

**Data sharing statement** No additional data are available.

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
