## [Reviewer comments · BMJ Open]

ARTICLE DETAILS

TITLE (PROVISIONAL)	Adapting to ulcerative colitis to try to live a 'normal' life: a qualitative study of patients' experiences in the Midlands region of England
AUTHORS	McMullan, Christel; Pinkney, Tom; Jones, Laura; Magill, Laura; Nepogodiev, Dmitri; Pathmakanthan, Shri; Cooney, Rachel; Mathers, Jonathan

VERSION 1 - REVIEW

REVIEWER	Dr Lesley Dibley University of Greenwich London UK
REVIEW RETURNED	17-May-2017

GENERAL COMMENTS	This is a very interesting paper which contributes meaningfully to the evidence base on the impact of this chronic illness (UC) on patients' lives, and what they do to try and adapt to it. It articulates clearly the changeable nature of adaptation over time and according to disease activity, and illustrates that this is not necessarily a one-way, linear process, or related to disease severity. The paper is well written, with clear and transparent description of processes. Tables and figures are useful. The work complements other emerging work in the field which is revealing the impact of IBD on patient's lives. There are only a few minor adjustments required: In the abstract, please insert 'for' in the sentence describing participants to read diagnosed with UC 'for' between 1 and 22 years; otherwise it reads as if the participants were diagnosed when they were between 1 and 22 years old. In the introduction, first sentence should read 'affects' the larger bowel, not 'effects' the large bowel. In the methods sections (Settings and study design), please add a sentence at the end of the paragraph to clarify the qualitative methodology used. It looks and reads like exploratory qualitative research (also sometimes labelled as qualitative description research) which has no specific underpinning philosophy but is useful for investigating patient experience and utilises expected qualitative methods. In the results section (Participants' descriptions of adaptation strategies and their intended purpose - prevent embarrassment), please reconsider the use of the word 'admitted' in relation to wearing nappies. This suggests a value judgement (of disapproval,
---

	or an expectation that this is not a good thing) and I recommend changing the word to 'reported' instead.
--	---

REVIEWER	Mark Löwenberg Academic Medical Center, Amsterdam, Netherlands
REVIEW RETURNED	29-May-2017

GENERAL COMMENTS	The aims of this study are (1) to provide a framework that is able to categorize whether patients are able to adapt to and lead a 'normal' life with UC and (2) to detail the factors that influence this. This manuscript presents valuable findings to provide insights in patients' experiences for healthcare professionals, however I have a few remarks:  - All patients included in this study were patients in remission and with a relatively mild form of UC, therefore no generalized conclusions can be drawn about all UC patients, since influence of disease activity and severity is not present. Therefore I suggest to add the words 'with mild to moderate UC in remission' instead of only 'UC' in the aim. - It is not clearly described what medical treatment these patients received at the time of the interviews. In the methods it is only stated that: 'patients were on standardized medical therapy' [line 126] and in the results it is stated that: 'all but one were on medical therapy, with three having taken biologic therapies' [lines 195-197]. I would also suggest to summarize this in table 2. - It is stated that 12 of the 28 participants could be categorized as experiencing some form of normal life with UC [line 211-212]. I suggest to clearly state how the authors defined normal life or type of normality at the time of the interview [table 2], since is an important outcome of the study. - Table 2: If additional information is available about social conditions of the participants, it would be interesting to add information regarding for example: marital status, number of children and employment status. - The conclusion that medication alone is most often insufficient to achieve a 'normal life' is too general [lines 546-548]. To my opinion statements about the influence of medical treatment cannot be generalized in this paper, since UC treatment was the same in the majority of the included patients. I suggest to adjust this conclusion to the scope of your study: mainly patients that did not use biologicals yet. We do not know if patients that use biologics would have come to the same conclusion.
--

VERSION 1 – AUTHOR RESPONSE

Reviewer 1

1) In the abstract, please insert 'for' in the sentence describing participants to read diagnosed with UC 'for' between 1 and 22 years; otherwise it reads as if the participants were diagnosed when they were between 1 and 22 years old.

Response: We have inserted 'for' in the sentence, as requested.

2) In the introduction, first sentence should read 'affects' the larger bowel, not 'effects' the large bowel.

Response: We have amended the typographical error.

3) In the methods sections (Settings and study design), please add a sentence at the end of the paragraph to clarify the qualitative methodology used. It looks and reads like exploratory qualitative research (also sometimes labelled as qualitative description research) which has no specific underpinning philosophy but is useful for investigating patient experience and utilises expected qualitative methods.

Response: We have added a sentence and further references at the end of “setting and study design” section within the methods section to make it clearer to the reader which type of qualitative methodology we used.

4) In the results section (Participants’ descriptions of adaptation strategies and their intended purpose - prevent embarrassment), please reconsider the use of the word ‘admitted’ in relation to wearing nappies. This suggests a value judgement (of disapproval, or an expectation that this is not a good thing) and I recommend changing the word to ‘reported’ instead.

Response: We acknowledge that the word ‘admitted’ could have a negative connotation and have changed it to ‘reported’

Reviewer 2

1) All patients included in this study were patients in remission and with a relatively mild form of UC, therefore no generalized conclusions can be drawn about all UC patients, since influence of disease activity and severity is not present. Therefore I suggest to add the words ‘with mild to moderate UC in remission’ instead of only ‘UC’ in the aim.

Response: We understand that we cannot generalise our conclusions to all UC patients. Whilst all patients were in remission at entry to the pilot trial, some of our patients did describe experiencing moderate/severe UC (see Table 2). Some of the patients included in our interview sample went on to have a colectomy soon after interview. We do not believe that our entire sample demonstrates a relatively mild form of UC, and as a result, we feel that we should maintain to our original aim.

2) It is not clearly described what medical treatment these patients received at the time of the interviews. In the methods it is only stated that: ‘patients were on standardized medical therapy’ [line 126] and in the results it is stated that: ‘all but one were on medical therapy, with three having taken biologic therapies’ [lines 195-197]. I would also suggest to summarize this in table 2.

Response: We believe the information requested on medication was already included in Table 2, under the ‘Medication’ column.

3) It is stated that 12 of the 28 participants could be categorized as experiencing some form of normal life with UC [line 211-212]. I suggest to clearly state how the authors defined normal life or type of normality at the time of the interview [table 2], since is an important outcome of the study.

Response: We believe a description of ‘normal’ (based on Sanderson et al., 2011) was provided in Table 1.

4) Table 2: If additional information is available about social conditions of the participants, it would be interesting to add information regarding for example: marital status, number of children and employment status.

Response: We have added information of marital and employment status to Table 2. We have not

included the number of children, as this was not available for all participants.

5) The conclusion that medication alone is most often insufficient to achieve a 'normal life' is too general [lines 546-548]. To my opinion statements about the influence of medical treatment cannot be generalized in this paper, since UC treatment was the same in the majority of the included patients. I suggest to adjust this conclusion to the scope of your study: mainly patients that did not use biologicals yet. We do not know if patients that use biologicals would have come to the same conclusion.

Response: We have amended the conclusion to reflect the fact that only three of our patients were using biologics.

VERSION 2 – REVIEW

REVIEWER	Mark Löwenberg Academic Medical Center, Amsterdam, the Netherlands
REVIEW RETURNED	27-Jun-2017

GENERAL COMMENTS	I agree with all responses of the authors. Although I still have 1 suggestion concerning question 2). The authors answered "We believe the information requested on medication was already included in Table 2, under the 'Medication' column." However, the only information the authors provide is: Yes, No or Biologics. Can you please specify this medication? For example: "5-aminosalicylates / 5-ASA"?
--

VERSION 2 – AUTHOR RESPONSE

Reviewer 2

1) I agree with all responses of the authors. Although I still have 1 suggestion concerning question 2). The authors answered "We believe the information requested on medication was already included in Table 2, under the 'Medication' column." However, the only information the authors provide is: Yes, No or Biologics. Can you please specify this medication? For example: "5-aminosalicylates / 5-ASA"?

Response: We have provided the medication regime the patients were under at the time of the interview (ie. 5-ASA, Immunomodulators, Biologics)